# SARS-CoV-2 transmission among health care workers, an outbreak investigation using whole-genome sequencing

K. S. te Paske[1☯]*, C. van Tienen[2‡], D Dunk[2,3‡], D. van Pelt[4‡], P. W. Smit[2,3☯]

**1** Department of Infectious Disease Control, Municipal Health Service Haaglanden, The Hague, the Netherlands, **2** Department of Medical Microbiology, Maasstad Hospital, Rotterdam, the Netherlands, **3** Molecular Diagnostic Unit, Maasstad Hospital, Rotterdam, the Netherlands, **4** Department of Occupational Health, Maasstad Hospital, Rotterdam, the Netherlands

☯ These authors contributed equally to this work.
‡ These authors also contributed equally to this work
* karlijntepaske@outlook.com

**Data Availability Statement:** On https://gisaid.org researchers can download all of the sequencing data that is used in this article. However one has to register (free). Our data is named as follows:

## Abstract

### Background

We report an outbreak investigation to map intra-hospital transmission among health care workers (HCW) using epidemiological and whole-genome sequencing data.

### Methods

Fourteen clinical wards (COVID-19 and non-COVID-19) with high infection rates of SARS-CoV-2 among HCW were selected and demographical, epidemiological and sequencing data were collected of all HCW testing positive by RT-PCR. Clustered cases were identified based on first disease onsets and differences in single nucleotide polymorphisms (SNP's) and were analysed for additional characteristics.

### Results

Data was collected for 123 HCW. Out of 123 HCW, 65 (53%) worked at eight non-COVID-19 wards, 56 (46%) at four COVID-19 wards, one (<1%) worked on several wards and for one (<1%) it was unknown. One major cluster (n = 34) and three minor clusters (n = 2,3,4; total n = 9) comprising of 43 HCW (35%) were found after comparing our study population (n = 123) with the circulating regional sequences (n = 819). In clustered cases work was most often the suspected source of infection and continuing work while having symptoms occurred in all clusters, ranging from 1–6 days.

### Conclusion

Our findings strongly indicate transmission of SARS-CoV-2 among HCW. Whole-genome sequencing is useful for identification of clusters and can give direction to targeted infection prevention measures.

"hCoV-19/Netherlands/ZH-MZ-1/2020" up until –>
"hCoV-19/Netherlands/ZH-MZ-123/2020".

**Funding:** The author(s) received no specific funding for this work.

**Competing interests:** The authors have declared that no competing interests exist.

## Background

Ever since the first reported case of COVID-19 in February 2020, the Netherlands is dealing with SARS-CoV-2 and its consequences. The second wave started in July followed by a steep rise in new daily cases up to 11.000 in December 2020 [1]. National public health measures were repeatedly intensified and the national testing capacity was constrained due to high demands [2]. Large numbers of COVID-19 infections among patients and health care workers (HCW) have posed pressure on all hospitals. Several studies indicate intra-hospital transmission between HCW [3,4]. In response to high infection rates in various clinical wards in one hospital in the Netherlands, we conducted an epidemiological and genomic outbreak investigation of hospital personnel with COVID-19 to gain insight in intra-hospital transmission.

## Methods

### Hospital context

This study was performed at one hospital in the Netherlands, during 4 September—31 December 2020. At the time, 6468 hospital employees were employed and 3724 RT-PCR tests were performed (personnel only) of which 2324 unique employees (S1 Table).

According to hospital policy, all HCW experiencing symptoms consistent with COVID-19 underwent oro-nasopharyngeal swab RT-PCR testing and were instructed to self-isolate. Awaiting test results, HCW were allowed to work with a surgical mask (type 2R) provided that symptoms were mild and their work was considered crucial (irreplaceable, indispensable and working remotely impossible) for continuity of patient care. In case HCW tested positive, they remained on sick leave until seven or fourteen days (depending on severity of clinical course) after disease onset and >24 hours free of symptoms. In case mild symptoms remained present, HCW resumed work after a positive SARS-CoV-2 antibodies test (Wantai, Beijing, China) at day 10. Personal protective equipment supplies and laboratory testing capacity for personnel were sufficient.

### Study design

Fourteen clinical wards (COVID-19 and non-COVID-19) with positive HCW were selected and all HCW working in these wards who tested positive for SARS-CoV-2 were enrolled. Departments with a high likeliness of receiving COVID-19 patients were registered as COVID-19 wards and the remaining departments as non-COVID-19 wards. The molecular diagnostic unit performed RT-PCR testing for SARS-CoV-2 and whole genome sequencing. The occupational health department inventoried demographical and epidemiological data of positive HCW. The Medical Research Ethics Committees United (MEC-U) waived the need for ethical approval as samples were collected for routine diagnostic care. Samples and data were processed anonymously to ensure privacy of our co-workers.

### Whole genome sequencing and phylogenetic analysis

RT-PCR was performed on a fast RT-PCR platform (NeumoDx, Qiagen, Germany). Samples with a cycle threshold below 30 were considered for sequencing. Oxford Nanopore MinION sequencing was performed following ARTICv3 (LoCost) protocol [5], using R9.04.01 flow cells on a MinIon Mk1b or Mk1C device. Basecalling and demultiplexing was performed using Guppy and further analysed using medaka for consensus and variant calling according to the ARTIC pipeline [6]. Phylogenetic analysis and data visualization was done in Pathogen.watch [7].

Since an epidemiological link combined with few single nucleotide polymorphisms (SNP's) is suggestive for nosocomial transmission and a higher number of nucleotide differences is indicative for acquisition from a different source, cluster criteria were met in cases with disease onsets within 14 days combined with a variable SNP variation (maximum of 5), bearing in mind that the mean evolutionary rate of all SARS-CoV-2 lineages is 4 nucleotide changes per month [8]. Commonalities in ward, position, source of infection and last working day between cases were considered as additional evidence.

SARS-CoV-2 sequences reported during September–December 2020, in the province in the Netherlands where the hospital is situated, were downloaded from GISAID and aligned with our data on Pathogen.watch. Solitary cases merging with the reference phylogenetic tree were assumed to be community acquired and marked as non-clustered cases. Cases standing out were considered as hospital acquired and marked as clustered cases. Pearson's chi-squared test was used for statistical analysis of group differences.

## Results

### Study population characteristics

A total of 197 HCW were selected for data collection of whom 123 (62%) were successfully sequenced. See Table 1 for study population characteristics. Out of 123 HCW, 65 (53%) worked at eight non-COVID-19 wards, 56 (46%) at four COVID-19 wards, one (<1%) worked on several wards and for one (<1%) it was unknown. Close patient contact was likely for 91 out of 123 (74%) HCW.

The study population (n = 123) was mostly representative for the complete hospital personnel testing positive (n = 662) (S1 Fig). Lineages detected most were B.1.221 (n = 46, 37%), B.1.36 (n = 35; 28%), B.1.177 (n = 16; 13%) and B.1.160 (n = 12; 10%), which corresponded with the most frequent circulating lineages in the concerning province in the Netherlands (n = 819).

### Cluster characteristics

Comparing our study population (n = 123) with the circulating regional sequences (n = 819), one major cluster (n = 34) and three minor clusters (n = 2,3,4; total n = 9) comprising of 43 HCW (35%) were found using the cluster definitions specified in the methods section. Features of clustered and non-clustered cases are described in Table 1. To provide a clear view, Fig 1 depicts the phylogenetic tree of only our study population. See S2 and S3 Figs in for the phylogenetic trees including circulating regional sequences.

**Table 1. Characteristics of clustered cases, non-clustered cases and study population (clustered and non-clustered combined).**

| | Clustered cases | Non-clustered cases | p-value | Study population |
|---|---|---|---|---|
| **Number of cases** | 43 | 80 | | 123 |
| **Median age (range) in years** | 32 (19–66) | 34 (19–65) | | 33 (19–66) |
| **HCW's suspected source of infection, location** | 30 (70%) work<br>5 (12%) private<br>8 (18%) unknown | 37 (46%) work<br>22 (28%) private<br>21 (26%) unknown | 0.09<br>0.07<br>0.4 | 67 (54%) work<br>27 (22%) private<br>29 (24%) unknown |
| **HCW's suspected source of infection, person** | 28 (65%) unknown<br>5 (12%) household<br>6 (14%) colleague<br>4 (9%) patient | 54 (67%) unknown<br>19 (24%) household<br>3 (4%) colleague<br>4 (5%) patient | 0.9<br>0.1<br>0.04<br>0.4 | 82 (67%) unknown<br>24 (20%) household<br>9 (7%) colleague<br>8 (6%) patient |
| **Number of HCW working with symptoms** | 11 (26%) | 15 (19%) | 0.4 | 26 (21%) |

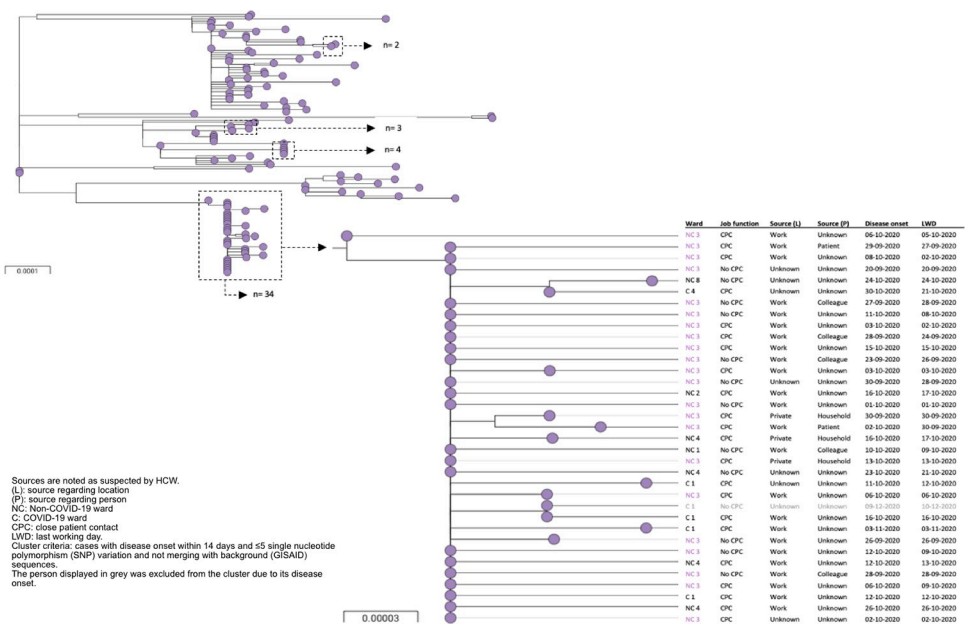

**Fig 1. Phylogenetic tree of study population (n = 123) with four clusters indicated.** The largest cluster is paired with additional information.

HCW from the major cluster (n = 34) worked in four non-COVID-19 wards (n = 28; 82%) and two COVID-19 wards (n = 6; 18%). This outbreak lasted 45 days reaching from September until November. Work (n = 24; 70%) was most often the suspected location of infection while the person whom they got infected by was mostly unknown (n = 24; 70%) (Fig 1). Having the exact same sequence (Fig 1), 22 (65%) HCW were likely to be infected at work and 7 (21%) continued working while having symptoms, ranging from 1–3 days.

HCW from the minor clusters combined (n = 2,3,4; total n = 9) worked in two COVID-19 wards (n = 6; 67%) and three non-COVID-19 wards (n = 3; 33%). Work (n = 6; 67%) was most often suspected as a source. Out of 9, 4 (44%) continued working with a range of 1–6 days.

## Discussion

This study used sequencing and epidemiological data to map intra-hospital transmission among a selection of HCW and disclosed one major and three minor clusters of SARS-CoV-2 involving both COVID-19 and non-COVID-19 wards.

In the study population, the majority reported to be infected at work (54%). It was mostly unknown who the exact person was (67%) and a minority stated it to be a colleague or patient (7% and 6% respectively) (Table 1). Maltezou et al. [9] found that colleagues were the source of exposure in 48% of the infected HCW, indicating the contribution of HCW-HCW transmission.

We also compared clustered cases with non-clustered cases. Clustered cases appointed colleagues more often as a source of infection compared to non-clustered cases (14% and 4%, p<0,04, Table 1). Piapan et al. [3] found that 71% of HCW used personal protective equipment (PPE) during working hours, but used PPE incorrectly during HCW meetings. This underlines the importance of stimulating HCW to adhere to preventative measures at all times and raising awareness of transmission risk in contact moments with colleagues.

Out of 123 HCW, 26 (21%) worked despite having symptoms. Hospital policy allowed HCW to continue working with symptoms under strict conditions. Even though there were no registered indications for staff shortages, it cannot be ruled out that departments resolved acute staff shortages themselves without reporting. Also, HCW could have been doubtful to call in sick with mild symptoms. Presenteeism (working despite being sick) can occur during outbreaks and is associated with mildness of symptoms. [10] Encouraging HCW to test when experiencing mild symptoms is essential for stopping transmission.

A strength of our study is that we collected demographical, epidemiological and genomic data of a large study population. Sequencing clearly defined a major cluster among four non-COVID-19 and two COVID-19 wards among the screened and sequenced HCW. Sequencing therefore highlighted at-work-transmission dynamics among several wards within a short timeframe.

However, this study also has limitations. The research team selected clinical wards with high infection rates among HCW, so clusters in other wards may have been missed and selection bias cannot be excluded. Information about HCW having contact with colleagues outside of work is lacking and epidemiological data was self-reported by HCW. Although hard to obtain otherwise, it could entail recall bias. Finally, patients were not included in our study whilst they could be the missing link. We attempted covering this by comparing our data to regional circulating sequences selected randomly at various sites to represent genetic diversity of SARS-CoV-2 strains in the Netherlands.

## Conclusion

Clustered cases of COVID-19 among HCW occur with indications of at-work-transmission. Sequencing can aid infection prevention efforts by conveying a clear message: the importance of adherence to standard precautions by HCW as keeping distance, performing hand hygiene at key moments and testing when experiencing symptoms, in contact with colleagues as well. Large interventions in infection prevention policy were not necessary, but there was a need to repeatedly remind the HCW. Educating and raising awareness of transmission risk among HCW remains important for prevention. In-depth interviews with HCW about risk perception, high risk contact situations and adherence to infection prevention measures could provide valuable information for targeted interventions.

## Supporting information

**S1 Fig. Number of daily tests among three populations of health care workers (HCW) in the hospital in the Netherlands, 4 September– 31 December 2020.**
(TIF)

**S2 Fig. Phylogenetic tree of study population (n = 123) and regional sequences in the Netherlands (n = 819), September–December 2020.** This figure depicts the total phylogenetic tree with the study population and regional sequences combined. Regional sequences of the concerning province are downloaded from GISAID.
(TIF)

**S3 Fig. Phylogenetic tree of study population (n = 123) with filtered out regional sequences in the Netherlands (n = 819), September–December 2020.** The purple circles represent the study population. The four clusters are indicated. Tree branches without purple circles represent filtered out regional sequences. Regional sequences of the concerning province are downloaded from GISAID.
(TIF)

**S1 Table. Diagnostic and demographic characteristics of three intra-hospital employee populations, 4 September– 31 December 2020.**
(TIFF)

## Acknowledgments

We would like to thank all employees of the hospital and laboratory where this research took place for their hard work in dealing with the COVID-19 outbreak.

## Author Contributions

**Conceptualization:** C. van Tienen, P. W. Smit.

**Data curation:** K. S. te Paske.

**Formal analysis:** K. S. te Paske.

**Investigation:** D Dunk, D. van Pelt.

**Project administration:** P. W. Smit.

**Resources:** D Dunk, D. van Pelt.

**Supervision:** P. W. Smit.

**Visualization:** K. S. te Paske.

**Writing – original draft:** K. S. te Paske.

**Writing – review & editing:** K. S. te Paske, C. van Tienen, D Dunk, D. van Pelt, P. W. Smit.

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
