## [Decision Letter · Decision Letter 0]

6 Mar 2023

SARS-CoV-2 transmission among health care workers, an outbreak investigation using whole-genome sequencing.

PONE-D-22-31851

Dear Dr. te Paske,

We’re pleased to inform you that your manuscript has been judged scientifically suitable for publication and will be formally accepted for publication once it meets all outstanding technical requirements.

Kind regards,

Kelli L. Barr, Ph.D.

Academic Editor

PLOS ONE

Journal Requirements:

1. Your ethics statement should only appear in the Methods section of your manuscript. If your ethics statement is written in any section besides the Methods, please delete it from any other section. 

Reviewers' comments:

Reviewer's Responses to Questions

**Comments to the Author**

1. Is the manuscript technically sound, and do the data support the conclusions?

Reviewer #1: Yes

Reviewer #2: Yes

2. Has the statistical analysis been performed appropriately and rigorously? 

Reviewer #1: N/A

Reviewer #2: Yes

3. Have the authors made all data underlying the findings in their manuscript fully available?

Reviewer #1: Yes

Reviewer #2: Yes

4. Is the manuscript presented in an intelligible fashion and written in standard English?

Reviewer #1: Yes

Reviewer #2: Yes

5. Review Comments to the Author

Reviewer #1: This is a neat and very straight forward outbreak investigation report, where the authors use epidemiological information combined with whole genome sequencing to investigate possible transmission routes of SARS-CoV-2 between healthcare workers in a hospital. The major limitation is that no patient samples are included, but a large number of community samples have been used to add a background for inerpretation of sequencing results and genereation of clusters. This important limitation has been properly adressed in the manuscript. There is no extensive discussion regarding the cut-off value of 5 SNPs to define a cluster, but the data presented support the results in a nice way.

Reviewer #2: Thank you for the opportunity to revise this manuscript, which deals with a pivotal public health issue, i.e. SARS-CoV-2 transmission among healthcare workers.

I found the study design appropriate, and the methods are clearly described.

The introduction section is clearly presented and the abstract is informative.

The discussion and conclusion are supported by the main findings of the research.

Overall, I enjoyed reading the manuscript and believe it will be of interest to the readers, as it falls within the aims and scopes of the Journal.

6. PLOS authors have the option to publish the peer review history of their article (what does this mean?). If published, this will include your full peer review and any attached files.

Reviewer #1: No

Reviewer #2: No

---

## [Editor Report · Acceptance letter]

23 Mar 2023

PONE-D-22-31851 

SARS-CoV-2 transmission among health care workers, an outbreak investigation using whole-genome sequencing. 

Dear Dr. te Paske:

I'm pleased to inform you that your manuscript has been deemed suitable for publication in PLOS ONE. Congratulations! Your manuscript is now with our production department. 

Kind regards, 

on behalf of

Dr. Kelli L. Barr 

Academic Editor

PLOS ONE